# A Simplified *A Priori* Theory Of Meaning
# –Nature based AI 'first principles'–

**Marcus Abundis**
Bön Informatics
Aarau, Switzerland
email: 55mrcs@gmail.com

## Abstract

This paper names **structural fundaments** in 'information', to cover an issue seen by Claude Shannon and Warren Weaver as a missing "theory of meaning". First, varied informatic roles are noted as likely elements for a general theory of meaning. Next, Shannon Signal Entropy as a likely "mother of all models" is deconstructed to note the signal literacy (logarithmic Subject-Object primitives) innate to 'scientific' views of information. It therein marks GENERAL intelligence 'first principles' and a dualist-triune (2-3) pattern. Lastly, it notes 'intelligence building' as named contexts wherein one details *meaningful* content—rendered via material trial-and-error—that we later extend abstractly. This paper thus tops today's vague sense of Open World 'agent intelligence' in artificial intelligence, framed herein as a multi-level Entropic/informatic continuum of 'functional degrees of freedom'; all as a mildly-modified view of Signal Entropy.

—Related video found at: *The Advent of Super-Intelligence*.

## 1 Introduction and Background: 'general information'

Many roles fill our eternally-dynamic simple-to-complex cosmos. One corner of that cosmos holds life where 'agents' adapt to **directly**-imposed selection forces via **indirect** 'referential' means or expire. Human agents notably adapt via indirect 'informatic abstraction' of direct events.[1] Here, agent INFORMATION is always 'about something', seeking to convey knowledge or intelligence about direct events, where better detail on 'How things work and fall apart' has more value/meaning. In this informatic enterprise, we make 'psychological artifacts' (ideas) into myriad material forms for better survival, using 'tools'—a process driving today's vast 'techno-cultural ecology'.

As humanity's main adaptive path, detailing that 'general informatic process'—onto today's artificial intelligence and beyond (AI/AGI)—is this paper's focus.

In grasping at a general 'scientific view' of information a key issue has been noted across disciplines, by varied individuals:
• "solving intelligence", Demiss Hassabis, Google Deep Mind (Burton-Hill, 2016),
• "de-risking science", Edward Boyden, MIT Media Lab (Boyden, 2016),
• "do submarines swim?", Edsger Dijkstra (1984), Eindhoven University, computer science,
• "symbol grounding problem", Stevan Harnad (1990), Université du Québec, cognitive science,
• "theory of meaning", Claude Shannon and Warren Weaver (1949), information theory, and more.

Each such 'gap' holds its own sense of the matter, but all can be seen as and reduced to one key informatic lapse. Shannon and Weaver were first to see this as a missing "theory of meaning" but it has since held many roles (as above). These 'gaps' arise due to a singular/universal statistical view of information in Claude Shannon's *A Mathematical Theory of Communication*, versus notions of information as 'meaningful/semantic content'. But, Shannon and Weaver (1949) soon saw *Theory of Communication* abuse (now being called 'information theory') would lead to "disappointing and bizarre" results, where a missing "real theory of meaning" (ToM) showed the theory being "bal-

---

[1] Informatic: energy-matter events as **direct** functions' (object interactions), each posing an 'agent chance' for sign/signal perception, creation, exchange, or processing, as **indirect** information' or data about *functions*.

looned to an importance beyond its actual accomplishments" with "an element of danger" (Shannon, 1956)—due to Signal Entropy's[2] "missing" *meaningful content* and odd "surprising" statistical role.

This paper offers a new approach. For example, deeper study shows differences in how we view *(S)ubject* information (raw percepts/data, object relations, 'qualia') and *(O)bject* information (self-evident matter, quantity, firm truths, etc.), hereafter (*S*) and (O), and S-O. Shannon (1948) used this split view to develop Signal Entropy, claiming "*[S]emantic aspects . . . are irrelevant to the [O] engineering problem*" (emphasis added) in order to isolate and model the latter role. If we accept this split view, (*S*) and (O) must also apply to a ToM, with (*S*)emantic aspects as a 'missing something'. But the terms (*S*)ubject and (O)bject are used variably, as are 'information' and 'intelligence', never truly detailed in relation to the other. They instead remain 'un-reconciled', driving the cognitive quagmire we have today (Dennett, 2013). With no uniform S-O base, myriad "disappointing and bizarre" (Eco, 2014; Shannon & Weaver, 1949) informatic notions instead abound—the central issue this paper targets, toward better focused AI and ensuing 'science'.

To better see an S-O split, mathematics seems 'purely objective' said to omit subjective roles from its arguments as an intellectual ideal (theoretical mathematics). But mathematics without *subjective elemental facts* as initial conditions (base data on 'primitives') is a "fact-free science" of little practical use (Feynman, 1964; Smith, 1995). Only if (*S*) and (O) roles are joined do predictive models arise as 'functionally reconciled' applied mathematics. If we look for other firm (O) views, the Standard Model of particle physics and Periodic Table are good candidates. But their 'objective success' often ignores that they arose from a line of *(S)ubjective elemental observations*, normalized (functionally reconciled) via experiment and peer review. Only after enough 'primitive evidence' was *subjectively discerned* and *subjectively named* by varied individuals, in many experiments, over decades, were models posited and *subjectively agreed* as being innately (O)bjective. Such *meaningful* '(*S*)teps' drive a GENERAL sense of 'informatic intelligence' or **functionally verified** S-O inter-relations.

Thus, the claim made here is that GENERAL/LOGICAL S-O (*S*)teps—detailed below—help correct 'gaps in meaning', and further support a sense of GENERAL INTELLIGENCE[3], along with informatic 'first principles' needed for firm AI/AGI/agent gains. But **Objectified**-*Subject* (**O**-*S*) roles like the Standard Model and Periodic Table are so accepted we forget their (*S*)ubjective origins. 'Objectivity' itself cannot even be implied if not first *subjectively sensed*, 'discovered' or 'imagined' by someone, before airing a 'sense-making' hypothesis. But GENERAL S-O (*S*)teps for framing new **O**-*S* roles (intelligence, meaning) are faint. In further studying S-O roles we see raw (*S*)-percepts (Stanley et al., 2017; Lewis, 1970) plainly precede **O**-*S* aims. I thus label this project **S-O modeling**[4]—want of a generative *uniform* S-O base, toward *diverse* intelligent/meaningful '(*S*)et' **O**-*S* roles.

Work of neuro-anthropologist Terrence Deacon (2013), biologist Stuart Kauffman (2024), and others (Wu, 2012; Thórisson & Talevi, 2024) mark early efforts at S-O modeling. Deacon's 'multi-state' view has Shannon's Signal Entropy, Boltzmann's thermodynamic entropy, and Darwin's evolution by means of natural selection (EvNS) as linked vistas (Dennett, 2013), with "structural, referential, and normative" facets (Deacon, 2020; 2024). This Shannon-Boltzmann-Darwin view suggests 'converged science' in a contiguous role, but its thermodynamic core omits wider physics-based models (Deacon, 2017) (four fundamental forces). Also, as the work is littered with neologisms and difficult prose (Dennett, 2013; Fodor, 2012; McGinn, 2012) it lacks breadth and clarity. Still, the strength of Deacon's *multi-state entropic study* is that it poses a bottom-up view (minimal logical gaps), is innately creative (re adaption), in a Natural contiguous role (key to any GENERAL theory, across domains), with 'simple-to-complex' functional ties (thermodynamic entropy, Signal Entropy, and EvNS). Beyond Deacon's view, Kauffman suggests Natural "order for free" and "adjacent possibilities", tied to Gibson's (1977) "affordance", as further *structural fundaments*.

---

[2]See Figures 1 and 2 for a brief description and discussion of Signal Entropy.

[3]**S-O modeling** covers detail missed by AI's typical "Intelligence measures an agent's ability to achieve goals in a wide range of environments" (Legg & Hutter, 2007). At issue, said *goals* and *environments* hold further (O)bjects/ives and (*S*)-ways/means, varied via trial-and-error in (*S*)electing 'functional Fits' and 'Fit-ness' (Müller, 2024), and in developing 'Open World' material reality, to make S-O modeling more foundational.

[4]Elsewhere I call **S-O modeling** Natural Informatics/Intelligence (NI) or 'thinking like Nature'.

## 2 Detailing an Informatic Base

As an alternative to Deacon's view, **S-O modeling** may seem paradoxic with 'opposed roles', per Shannon above. In 1901 Bertrand Russell saw a like clash in Georg Cantor's mathematical Set Theory, later called Russell's paradox. His solution asked that we see different 'types' of data exist—a cognitive advance that gave rise to Type Theory. S-O modeling demands a like advance in (*S*) and (O) 'data types' with "differed" (Bateson, 1979) "levels of abstraction" (Korzybski, 2010), to map otherwise paradoxic diverse-but-uniform and simple-to-complex roles. The problem is that Shannon information theory offers no systemic/informatic 'types' beyond (O) Signal Entropy as a single scalable statistical 'scientific role', without presumably-opposed (*S*)emantic aspects.

But (*S*)emantic aspects abound as 'metadata' with "differed" *meaning* in every formula, recipe, sheet music, blueprint and more, all detailing *types* of meaningful intelligence—that also map 'How things GENERALLY work' for diverse DISTINCT domains. For example, a Periodic Table atomic metadata **context** holds 'a type' of knowledge-about-data[5]—'material primitive' **content** of electron-neutron-proton triads as 92 Natural **O**-(*S*)et elements. Maps use 'symbolic primitive' **O**-*S* legends to detail map content. Even Assyrian clay tablets (3 kya) and Rosetta Stone note details *about* 'other tablets'. Myriad meaningful (*S*) metadata examples exist, mirroring (O) Signal Entropy's wide use, with both seen as diverse **O**-(*S*)et roles in the above examples—that also support AI/AGI agent aims, where (*S*) and (O) 'informatic atoms' underlie all aspects of actual **O**-*S meaningful intelligence*.

Beyond metadata, a unifying 'Meta-meta type' also exists. For example, with the Standard Model and Periodic Table holding diverse **O**-*S* meta-content, a view linking the Standard Model *with* the Periodic Table, chemistry, genomics, etc. evokes a contiguous Meta type (... O-*S*-O-*S* ...). Meta-meta 'logical primitives' mark GENERAL LOGIC across DISTINCT 'material primitives'—mapping 'How things GENERALLY work' across domains. If not for **domain-distinct** material/symbolic primitives (meta), linked via **domain-neutral** logical primitives (Meta), a ToM would be hopeless. Linked meta-to-Meta (*S*)teps echo Deacon's 'converged science' and the called for diverse-but-uniform and simple-to-complex S-O modeling base.

An early Meta-meta example is 'dialectics': thesis + anti-thesis = synthesis—seen across history in every technical advance and cultural leap. Darwin's *uniform* view of *diverse* evolving species is also Meta-meta. Type Theory is another Meta-meta (Type-of-types) example. Lastly, Signal Entropy is Meta-meta, supporting all of information technology and fitting so many domains that we call it "the mother of all models" (Hollnagel & Woods, 2005). Meta-meta shows GENERAL LOGIC for diverse DISTINCT material and symbolic primitives/roles/types/(*S*)teps—often as core scientific models, as with the above examples. Biologist Gregory Bateson (1979) called this Meta-meta *structural link* a "necessary unity" and a "pattern that connects" the cosmos, while science targets a kindred 'unified field theory' (UFT) and others aim to "mine a computational universe" (Wolfram, 2017).

A ToM thus contiguously maps GENERAL ⇄ DISTINCT 'data types'—meta-to-Meta—to show GENERAL *informatic processing* as humanity's 'main adaptive path' for AI/AGI/agency and beyond.

### 2.1 Glossary for a GENERAL Theory of Meaning—initial 'first principles'

The above names many (*S*) and (O) roles, types, and (*S*)teps, with a ToM targeting a Meta-meta *uniform* view of *diverse* roles. Before proceeding further, I detail some key informatic aspects. Foremost:

- '**Function**' is the term that best notes that 'Things GENERALLY work and fall apart', we hold as abstract *meaning*. **Functional** "affordance" is thus what we aim to *understand* as 'intelligence'. Here, O-*S*-O detail minimal **O**-(*S*)et functions (meta, 'simple'), and ... O-*S*-O-*S* ... mark varied contiguous dynamic simple-to-complex (Meta, evolving cosmos) functioning.
- **Adaptive** *Fit-ness*—after **O**-(*S*)et functions, agents seek **adaptive options** that abide, exploit, or otherwise 'Fit' a dynamic simple-to-complex cosmos; contra dynamic 'noise' as *extinction risk*.

As further *meaningful* types:

- **(*S*)ubject** and **(O)bject** are joint GENERAL *logical primitives*. Signal Entropy's S-O split implies dualism. Stated simply, (*S*) is 'relational joining' in (O)s as O-*S*-O or **O**-(*S*)et 'Fits'. A ToM thus

---

[5]Metadata: often defined as 'data about data', which ignores Russell's paradox in not naming 'data types' and simple-to-complex 'Levels'—but that are detailed herein as (*S*) and (O) (*S*)teps central to S-O modeling.

maps contiguous **O**-(*S*)et 'functional Fits' and (*v*)ariants (...O-*S*-O-$S^v$-$O^v$-$S^{v1}$-$O^{v1}$...) with Bateson-like "differed" S-O types and (*S*)teps, alongside Korzybski's "levels of abstraction". Next,

- **Metadata** is a 'well-ordered' DISTINCT role—each **O**-*S* Group marks a 'domain specific' *context* of (*S*)imilarly-named primitive material/symbolic functional *content*. Many such DISTINCT **O**-(*S*)et Group *contexts* fill our eternally dynamic cosmos.
- **Meta-meta** is a GENERAL role as 'domain-neutral' *uniform* LOGIC for *diverse* (*S*)et Groups. Meta-meta cross-maps **O**-(*S*)et domains and vaiants, via shifting (*S*) and (O) 'informatic atoms'.
- **Level** (*S*)teps—of DISTINCT simple-to-complex Groups, acute-to-slight (*S*)hifts in the underlying energy-matter/S-O admix drive "differed" 'functional degrees of freedom' (DoF). Here, abrupt shifts can yield DISTINCT *emergent* Level Fits. S-O 'informatic atom' (*S*)hifts thus serve as a GENERAL proxy for empiric energy-matter DoF (*S*)hifts and (*S*)teps .
- **Context** and **Content** in prior roles—(*S*) Group *contexts* with **O**-(*S*)et primitive *content*. Named contexts **with** firmly detailed content, alongside material/logical (*S*)teps, allow us to map simple-to-complex material reality, detailed as contiguous ToM functional segments.

The above mark 'types of meaning', where 'gaps' next show as:

- **Raw (*S*)-percepts** are a *meaningless type*—'things' we say exist but with faint (*S*)ense-making detail, e.g., Life, genomics, evolution, gravity. For meaning agents gather-and-interpret more (*S*) data in posited functions, with (*S*) and (O) trial-and-error ⇒ **O**-*S* metadata as the root of all *active intelligence* (applied/generative agent logic, and ensuing 'material proof').
- **Voids** are a next *meaningless type*—things we imagine exist but fail to grasp (dark matter, dark energy, quantum mechanics, etc.), and unnamed things we are wholly blind to, failing to (*S*)ense them in any useful way. Lastly,
- **Meaningless** roles are a *dysfunctional* 'functional type'—everything has proto-meaning, even noise/ignorance/absence, but can seem meaningless in differed contexts.

## 3 INITIAL TOM/S-O MODELING

The above initiates an **S-O modeling/**ToM base, with simple-to-complex (*S*)teps over one-step statistical vistas. Even with Bateson's unified sense, varied coevol generative (*S*)hifts must be named. Shannon and Weaver likewise saw three Levels (A, B, and C)[6] of needed study, with more to come, while Korzybski noted many "levels of abstraction". This all evokes mixed representational and computational challenges, requiring some manner of *multi-state analysis* (re Deacon). Lastly, Shannon and Weaver also saw that Signal Entropy's must be improved upon—causing Shannon to warn against using Signal Entropy as a general model (Shannon, 1956), leaving us to ask 'What way forward?'

### 3.1 SIGNAL ENTROPY AS GENERAL 'MULTI-STATE' S-O (*S*)IGNS

To answer 'What way forward?' I show Signal Entropy—a firm Meta-meta model—in a mildly altered *meaningful*-to-*meaningless* (multi-state) role: a max-possible functional span, with min-possible (*S*) and (O) steps (Figure 1); for an initial S-O modeling example. But first I clarify Shannon's claim "[S]emantic aspects ... are irrelevant to the [O] engineering problem", implying engineers are blind to (*S*)emantic or *meaningful* aspects.

Foremost, Signal Entropy itself defies Shannon's 'pure (O) claim' as engineers plainly pursue (*S*)emantic studies to make 'an engineer' an *Engineer*. Signal Entropy's empiric (*S*)emantics show in: Figure 1's *Engineered* (*S*)igns (number systems, alphabets, etc.), and an *Engineered* (*S*) logarithmic Fit in all messages—joint (*S*) GENERAL logic/rules. Such 'empiric Engineered roles' echo Deacon's *meaningful* "structural information"[7]. The problem with Shannon's (O) claim is that it mixes *levels of abstraction*, that Korzybski (2010) warned against, to assert '(O)bjective purity'. In ignoring its own (*S*)tructural *empiric meaning* and "adjacent possibilities", Signal Entropy falsely asserts "irrelevant semantics". But such simplified 'objective' views typify much of science, swap-

---

[6]Shannon's Level A covers Signal Entropy as "the technical problem", Level B marks "the semantic problem", and Level C marks "the effectiveness problem", but Shannon and Weaver (1949) never details B or C.

[7]Here, *structural information* broadly refers to 'specific (*S*) **and** (O) Fits', or a material event as 'some force applied to some object'[1], all of which a ToM ultimately maps.

ping Natural Open World complexity for more-workable (partial, segmented) 'closed' or 'isolated' system views. Ensuing 'logical fragments' offer *some* gain in mapping truly contiguous dynamic simple-to-complex Nature, but again, with prior-noted 'gaps'.

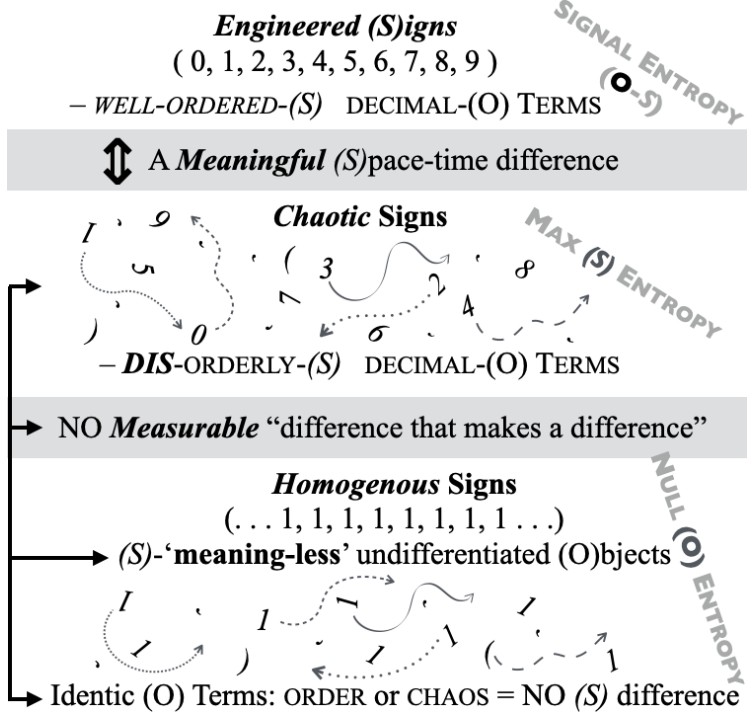

Figure 1: Engineered/Empiric (*S*)igns, Entropic (*S*)teps, and Min/Max ToM (*S*)tructure. Symbolic decimal primitives (top) show a *meaningful* agent-agreed **O**-(*S*)yntax as firm (*S*)pace-time Fits/order. All messages embody such Signal Entropy. Next, Chaotic Signs lack firm (*S*)pace-time Fits that messages require, shown as Max (*S*) Entropy (re thermodynamic entropy). Identic (O) signs (bottom) also bar messaging due to (O)-indistinct (*S*)pace-time Fits or Null (O) Entropy. Not shown is Null (*S*) Entropy as a collapsed (O) 'singularity', or *fully-un*"differentiated" S-O Fits. Lastly, Min (O) Entropy is (*S*)et in the Standard Model producing 92 initial Natural atomic elements/Fits. All roles can be shown via 'statistical mechanics', but which itself ignores *meaning-ful/less* aspects.

A ToM is thus wholly contiguous: simple-to-complex *open* material functions with *generative* Natural **and** Engineered *meaning*, alongside presumed-meaningless roles. Hence, Figure 1 shows one *meaningful* (**O**-*S*) and two *meaningless* 'at the limit' [MAX (*S*) and NULL (O)] roles as a widest-likely *meaningful*-to-*meaningless* (multi-state) vista, or 'How things GENERALLY work and fall apart' across an empiric cosmos, to initiate a ToM map we later enlarge with (S) and (O)/DoF detail.

In sum, Engineered 'logarithmic (*S*)igns' have *empiric meaning*, contra *meaningless* Max (*S*) 'noise' and Null (O) entropy—**minimal** *uniform* (*S*) and (O) 'informatic atoms', in **maximal**-*differed* DoF (*S*)teps—for a contiguous **proto-ToM**. Also, S-O duality with 'three GENERAL Entropic types' (all in Figure 1) mark a *uniform* dualist-triune (2-3) GENERAL pattern, where (*S*)pace-time has similar 2-3 traits[8]. I call this 2-3 vista simply 'Entropy' (generic expansion), with myriad material-and-symbolic, *orderly* and *dis-orderly*, min/max DoF (*S*)teps. But this view also lacks 'generative detail'—specific energy-matter admix shifts enacting DISTINCT Level (*S*)teps (see Glossary).

## 3.2 SIGNAL ENTROPY AS A GENERAL 'GENERATIVE ROLE'

Next, S-O modeling's generative detail shows in Signal Entropy's logarithmic base. For example, imagine a 3-term alphabet (A, C, and T) where all messages also hold 3 terms (Figure 2). Here, En-

---

[8]Dual-three-part (*S*)pace-time as SPACE with: height, width, and ***depth*** (2d simple/***3d realism***); and TIME with: past, future, and ***present*** (2d imaginal/***3d realism***), shown here as further-*nested* 2-3-**2** Fits.

gineered $X^n$ Fits suggest an 'empiric event' (logarithmic O-*S*-O joining) causing[9] 27 ($3^3$) message (*v*)-options, as a simple-to-complex S-O (*S*)tep: A, C, and T $\Rightarrow$ CAT, ACT, etc. Signal Entropy thus 'generates' as (*v*)aried (*S*) and (O) empiric Fits (S-O-*v*)—or $X^n$ as *infinitely-generative* "adjacent possible" simple-to-complex (*S*)teps. This S-O logarithm echoes genomic code creation/mutation in diverse species, and 'ln(*x*)' as "one of the most useful functions in mathematics, with applications throughout the physical and biological sciences" (Murray, 2024). Lastly, $X^n$'s 'causal structure' again holds a 2-3 pattern: $X^n$ = base-$X$ and term$^n$ (*S*)hifts, or (S-O-*v*) generative joining. But the problem now is that Signal Entropy is 'blindly generative', it omits (*S*) *meaning* beyond statistical (O)s—the issue raised above contra Shannon's pure-(O) claim, and that a ToM ultimately corrects.

| **CAT** | CCT | CTT | CCA | CAA | CTA | CTC | CAC | CCC |
|---------|-----|-----|-----|-----|-----|-----|-----|-----|
| **ACT** | AAT | ATT | AAC | ACC | ATC | ATA | ACA | AAA |
| *TAC* | *TAT* | TCC | TTA | TAA | TCA | TTC | TCT | TTT |

Figure 2: Scale-able/Select-able Signal Entropy. C, A, and T in *orderly* (*S*)pace-time Fits: an S-O volume with $3^3$ Signal Entropy (27 DoF). CAT and ACT are (*S*)et English words, "what you *do* say", others are (*S*)elect-able DoF options "what you *could* say" (Shannon & Weaver, 1949) (emphasis added). For example *TAC* and *TAT* hold 'meaning' in differed contexts (French, Old English, German). But all require 'agent agreement' on *meaning*: inter-(*S*)ubjective (O)bject operation as encultured **O**-*S adaptive functioning*. Without agreed **O**-(*S*)et functions, only 'informatic noise' arises. Lastly, shifting $X^n$'s root values makes S-O modeling infinitely +/- (*S*)cale-able and (*S*)elect-able.

*(S)ignal Entropy*'s now-noted '*meaningful* S-O' and '*generative* $X^n$' again suggest a **proto-ToM** of contiguous-scalable (gradual-to-acute) DoF—an empiric 'How things GENERALLY WORK' in the cosmos (Figure 2). Thus, so far we see: a) **O**-(*S*)igns as (*S*)et O-*S*-O functioning, b) contiguous-expansive . . . O-*S*-O-*S* . . . functioning, c) *meaningless* (*S*) and (O) 'noise' (Figure 1), and d) (*S*)tep-wise *generative* S-O-(*v*)ariability, all with e) Entropic 2-3 "structural information"—as a MEANING-FUL "mother of all models". But *now* the issue is that this is 'blind': it omits *meaningful* (*S*)election (Natural Fit-ness, 'reinforcement', EvNS) amid myriad **O**-*S* and S-O-*v* options.

Generating S-O options (Figure 2's $X^n$) innately begs/"affords" adaption. But such 'big data' vistas alone cannot treat 'gaps in meaning' due to unsure utility—not all options equally enact functions (Kauffman, 2024). For example, Figure 2 holds many S-O options that lack utility, akin to 98% of DNA thought to be 'non-coding' and many 'non-beneficial' mutations. Functionally "differed" *effective*-and-*efficient* roles require (*S*)election (trial-and-error, DISTINCT use) as specific contexts generatively-rendering meaningful content. Here, each trial yields more DoF detail/understanding of material "affordance" (*empiric* intelligence). A *uniform* aim of (*S*)tructural (*S*)urvival, with *diverse* material Fits, frames "order for free" Natural Rules for all matter and agents. Thus, I now cover (*S*)elective (*S*)urvival as a last (*S*)tep in ToM/S-O modeling.

## 4 (*S*)ELECTIVE/(*S*)TRUCTURAL SURVIVAL AND INTELLIGENCE

*(S)ignal Entropy*'s now-detailed *meaning* would seem to also support a statistical 'big data' view in its "surprising" statistics. But this ignores "contrast[s] between mental models that rely on statistical correlations and those that rely on causal mechanisms" (Mitchell & Krakauer, 2023). Without empiric/functional (*S*)election, 'big data' vistas alone omit a key scientific role (Wilson, 2009)—as a "fact-free science": eluding EvNS as a 'black box' sans 'first principles', with unsure 'primitives' and 'aims'. Big data *training* (trials) may improve the case some, but not with true scientific *structural clarity*. Shannon similarly saw miss-labeled 'information theory' as "disappointing and bizarre" and "ballooned to an importance beyond its actual accomplishments" with "an element of danger" (Shannon, 1956), where some now see the same in today's statistical AI (Mitchell, 2024).

---

[9]A 'logarithmic *tool*' applied by human hands, but also seen in Nature as *the natural logarithm* 'ln(*x*)'.

Still, modeling (S)election can be tricky. First, (S)election amid many GENERAL options **makes** DISTINCT roles, as varied functional Fits in diverse niches. But such mixed effects raise yet another issue—no 'one Fit for *all things*' exists, as widely (*v*)aried functional S-O 'Fit-ness' (S-O-*v*). For example, no one (S)-role for all (O)-carbon atoms exists, with differed ions and isotopes[10], no 'one way to be Human' exists in varied environs, no 'one type of fish', nor even 'one form of Lever or Wheel' exists. *(S)*election also holds *(s)*pecific *embodied structures* of ambiguous aspects— 'human hands' a notable example. Anaxagoras and Aristotle saw hands as an "instrument of all instruments" (Kirk et al., 1983; Maurette, 2017), with advent of 'six simple machines'. Kaufmann (2024) likewise saw many uses for screwdrivers. Such multi-use/multi-state GENERAL tools **and** diverse DISTINCT purpose-built roles mark 'computational multiplicity'[11]. Nature thus 'reductively-expands' (paradox) as simple-to-complex *(S)*elected DIVERSE-DISTINCT materials and *(s)*-agents, with shifting evolutionary trees, etc. where not all things are equally functional—Fit-***ness***.

As such, EvNS has (S)tructural *(s⇌S)*election of functions (*empiric* "affordance"), apart from statistical vistas. Here, chaos theory and the like offer some *structural insights*, while a ToM specifically targets *structural detail*. For example, if science 'describes-and-explains, cause-and-effect, in measurable-and-repeatable ways, with necessary-and-sufficient detail' a ToM is mostly descriptive, as is Signal Entropy. Both hold *meaningful* 'descriptive detail' **about** 'information'—Shannon's universal $X^n$ singular (O) statistics, versus *contiguous-dynamic* DoF as GENERAL/DISTINCT logical **O**-(S)ets and (S)teps. A ToM's mapped *(s⇌S)*tructural 'descriptive detail' thus meets Shannon and Weaver's call to improve Signal Entropy, using (S) and (O) informatic atoms to show 'gaps' alongside DIVERSE-DISTINCT and GENERAL functional (*v*)ariants.

'Science' also targets *structural detail*, that we (*v*)ariably expand. For example, the Standard Model suggests a latent UFT with partly-detailed gravity, dark energy, dark matter, and neutrinos as 'gaps'. And even if "[c]racks are beginning to show in the periodic table," (Powell, January 19, 2016) it still suitably details a *(S)*elected "order for free" 92 Natural elements, where we now *(s)*ynthesize new #93 – #118 elements. This synthetic '*(s⇌S)*uper-intelligent'[12] (SI) expansion of 'elements' divides GENERAL INTELLIGENCE (Natural 'GI' Rules) from human SI where fluid *(s⇌S)*elective '*dialectic adaption*' presents a Ground of Being (Bliss & Trogdon, 2021) for all things we call intelligent.

## 4.1 INTELLIGENCE BUILDING—TRIAL-AND-ERROR *(s⇌S)*ELECTION

Hence, *(s⇌S)elective* dialectic structure yields '*surprising* types of intelligence'. To see 'How and why?' DIVERSE-DISTINCT *(s⇌S)*elective intelligences arise, I return to computational multiplicity.

Our *contiguous-dynamic simple-to-complex* cosmos has many functions. From odd quantum facets, to purely-mechanical cause-effect, onto bio-logic stimulus-*processing*-response—many "order for free" *(S)tructures* exist. Many types of *(s)-intelligence* thus ensue via: niche diversity ⇒ agent diversity ⇒ intelligence diversity, as *(s⇌S)*urvival. Here, mixed agent outputs would seem to bar further *(s⇌S)*tructural detail, or measurable-and-repeatable *Patterns*. But 2-3-*2* roles show again as: a) dualist Life-Death, with three-part Natural *divisive*, *directive*, **purifying** (-/**+**) (*S*)election effects; upon b) land, water, air (O)-environs, contested via c) *(S)*-Life as *bacteria*, *archaea*, and **eukarya** (*no cell nucleus*/**nucleus**); along with *(s)* 'informatic *paths*' of genomic instincts, intuitive/proto-cognitive bio-mechanics/chemistry, onto full agent cognition—all as mixed/serial . . . O-*S*-O-*S* . . . roles/trials.

'Intelligence building' *content* next arises as one learns to walk, swim, fly, etc.—early *(s⇌S)*elective trial-and-error. From the ground up, genomics bestow each agent: 1) 'a body' [(O) material], amid 2) material niches [(O) environs], where 3) (*v*)aried *(s)*-agent acts yield *(S)*urvival/Death—a *material* **context** *(s⇌S)*electively rendering *meaningful* **content**—(O-*S*-O). Agents not acting 'effectively-and-efficiently' expire, while survivors continue in a serial 'reproductive' manner. It matters not what informatic path applies: genomic instincts/emotions, bio-mechanical/chemical intuitive 'think feeling', or full cognition, as all rely on '*indirect*/referential abstraction of *direct* material events'. All also use trial-and-error as *feedback*, where each trial poses more +/- DoFs in 'understanding' ma-

---

[10] Carbon has 14 isotopes, "unique among the elements in its ability to form strongly bonded chains" (of Chemistry, 2024), giving carbon a broad profile in forming material reality.

[11] Stephen Wolfram (2016) instead points to a sense of 'computational irreducibility'.

[12] Humans employing Natural Rules with supra-Natural effects reflects SUPER INTELLIGENCE (SI) (e.g., a ball-point pen). True GENERAL INTELLIGENCE (GI) implies a full grasp of Natural Rules (no 'gaps'). Human-Level intelligence (HLI) holds mixed-shifting *partial* GI and SI, which makes HLI unsuitable for modeling.

terial "affordance"—or '(*S*) and (O) trial-and-error ⇒ **O**-*S* metadata' as THE core of intelligence. We may abstractly extend this 'process' in many ways, but none that surpass '*verified* materiality'.

As an alternative account: *(S)* and *(s⇄S)* **functional** *material* "affordance" ⇒ **functional *(S)*** and *(s⇄S)* *material* trial-and-error ⇒ material 'proof' as *(s⇄S)*urvival ⇒ *informatic* **functional** understanding/*intelligence*; with *(S)* as Nature and *(s⇄S)* as *dialectic* 'agency'. This base 'rendering of *meaningful content*' thus offers "a framework for [grasping] what intelligence is" and is not (Mitchell, 2017), but requires we also grasp 'trial-and-error'. Earlier I note an 'informatic path': genomic instinct/emotions, intuitive bio-mechanics/chemistry, and full agent cognition. Next, the 'generative Life force' **animating** that path drives an agent's adaptive impulse and "breathes fire into the equations" (Hawking, 1988). The main *emotive forces*[13] involved here are: 1) discontent ('something must change!'), 2) curiosity ('open to explore') and, 3) imagination (abstract-recombinant 'visual/material ideation'), with countless DoF/trial-and-error/psycho-logical implications.

Mixed Fear/Hope DoF **with** mixed sensoria DoF *(s⇄S)*ustain all agents, well before '*(s)*cientific views' arise. We thus counter Natural *(S)* computational multiplicity with emotive *(s)*-informatic multiplicity, *(s⇄S)*electively refined by trial-and-error. Much later we develop ever-more-refined but still-abstract science and engineering, but always subject to materially verified *trial-and-error*.

### 4.2 A SIMPLISTIC ToM EXAMPLE

*Emotive* trial-and-error hinders further analysis, being psycho-logical in tone. But as *(s⇄S)*election is ultimately mediated via shifting direct 'fundamental energy-matter admixes'—(*v*)aried agent DoF **contra** Nature's shifting DoF, as 'adaption'—ensuing direct energy-matter effects allow more ToM mapping. For example, regard the Standard Model's 2-3 *proton*, *neutron*, and **electron** DoF, onto the Periodic Table's atomic elements—a *fundamental* expansive GENERAL-to-DISTINCT (Standard Model ⇒ Periodic Table) domain *context*/Level shift. Here, a specific energy-matter admix yields 92 elements: 1) strong nuclear attraction in nucleons ($10^{38}$ relative strength), **contra** 2) electromagnetic repulsion in protons ($10^{36}$ relative strength), and 3) an electron cloud, mark 2-3 **generative** DoF with *strong⇄electromagnetic* 'force impedance' as a *contested emergent* effect[14]—an *(x⇄X)* dialectic.

Ninety-two elements arise in force "differences": strong force's range is near the radius of a nucleon, but electromagnetic force has no like limit. Even if strong nuclear attraction is 100 times greater, electromagnetic repulse builds with more protons, until topping strong force limits (100x ≈ 92 protons). Here, large atoms start to falter and even larger atoms become impossible. After **this** *fundamental* GENERAL-to-DISTINCT *generative* shift, a **next** 'periodic table ⇒ chemistry' domain/*context* shift arises, via molecular bonds (*electromagnetic*, *covalent*, and **metallic**). This **next** 2-3 **generative** (*S*)tep holds myriad *emergent* 'molecular' DoF, where detailing molecule 'types' is **yet another** *emergent* generative task [another 'gap' (Powell, January 19, 2016)]. This short *(x⇄X)* example shows already known generative DISTINCT ToM (*S*)election as "order for free", while also alluding to 'gaps' (gravity, dark energy, dark matter, a cause for Life, etc.) awaiting reconciliation as a full *Meta-to-meta* ToM. Thus, much work extending this example to other Levels lies ahead.

## 5 CONCLUDING DISCUSSION

From the ground up, this *Meta-meta* "structural" account departs from usual scientific 'gaps' amid domains, and vague AI/AGI views of intelligence[3]. It notes 'How things GENERALLY work and fall apart' in the cosmos, with multiple (simple-to-complex) Levels of *informatic intelligence*, based on material/functional analysis and 'understanding'—beyond AI's statistical 'next term prediction'.

As such, likely ToM benefits are:
- a crude *force-driven* account of the cosmos with 'knowledge and gaps' toward eventual full Natural Intelligence (NI).
- a fluidly-scalable trans-disciplinary modeling tool, requiring minimal initial (*S*) and (O) input,
- where one variably 'zooms in and out' for better-focused/directed/dynamic analysis,
- elimination of AI 'black box' issues due to extensive 'mapping',
- named 'first principles' for all *intelligent projects* as AI, AGI, SI, GI, NI, HLI activity,

---

[13] Akin to four fundamental forces serially driving the Standard Model toward serial complex effects.

[14] Emergent: a *new* function unseen in prior/'lower' Levels.

- an observed contiguous dynamic GENERAL 2-3-**2** *nested* 'cosmic pattern'.
- a framework for other simple-to-complex GI vistas, to improve scientific/computational modeling, toward 'science' sans logical gaps and the "de-risking of science",
- tops 'anthropic/narcissistic impulses and biases', for true GI and 'common sense', and lastly,
- suggests an 'insight engine', with re-framed 'gaps', affording many likely eureka moments.

Despite likely ToM benefits, this merely poses 'a tool' for informatic/intelligent exploration—where **application** of that tool ultimately drives future gains with said application demanding much more S-O modeling work, toward a full ToM and Human NI vistas.

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

Kun Wu. The essence, classification and quality of the different grades of information. *Information*, 3, September 2012.

## A   APPENDIX

You may include other additional sections here.

