# OpenReview forum: "A Simplified a priori Theory of Meaning; Nature Based AI 'First Principles'"
_ICLR.cc/2025/Workshop/AgenticAI — ICLR 2025 Workshop AgenticAI Poster_

### Official Review · Reviewer_wTqX · 2025-03-03
**A Simplified a priori Theory of Meaning; Nature Based AI 'First Principles'**

**Rating:** 7
**Confidence:** 4

**Review:**

# 1. Summary

This paper attempts to complete Shannon’s information theory by distinguishing between the  subjective (S) and objective (O) aspects of information. It uses S-O modeling and Signal  Entropy to explain how meaning is created from data, in order to found a general theory of intelligence.

 # 2. Strengths

 1.  **Novel Insight:**
 The paper presents a new perspective by combining information theory with concepts from cognitive science.
2.  **Interdisciplinary:**
 It builds on the work of several disciplines to create a unified framework.
3.  **Conceptually  Rich:**
 It presents fairly conceptual and very interesting ideas on the production of meaning.

 # 3. Weakness
1. **Dense Language:**
   The manuscript is filled with many technical terms that make it hard to understand, especially for readers not familiar with the field.
2. **Limited Practical Validation:**
   The paper doesn’t include enough real-world examples or case studies, which makes it harder to see how the theory would work in practice.

---

### Official Review · Reviewer_4C38 · 2025-03-04
**A Simplified a priori Theory of Meaning; Nature Based AI 'First Principles'**

**Rating:** 3
**Confidence:** 3

**Review:**

### **Strength**:

1) **Strong Motivation**: The paper tackles an important problem in information theory and AI, namely the absence of a clear theory of meaning. It refers to a framework that explains how language, symbols, or information carries semantic content - essentially, how meaning is created, represented, and understood.  The authors are trying to develop a framework that explains how meaning emerges in both natural and artificial systems, which they argue is essential for advancing general artificial intelligence beyond purely statistical approaches.

2) **Interdisciplinary approach**: The paper draws from information theory, cognitive science, and philosophy attempting to synthesize concepts across disciplines.

### **Weakness:**

1) **Lack of clear research question**: The paper does not explicitly define a central research question or hypothesis. Instead, it discusses various broad and abstract ideas related to information theory and intelligence without a focused inquiry. Despite proposing a theoretical framework, the paper lacks formalism or precise definitions that would make its claims testable or falsifiable.

2) **Complex writing and paper structure**: The writing style is unnecessarily complex and philosophical, employing convoluted sentences that impede comprehension rather than facilitating it. Many key concepts, arguments, and claims remain vaguely defined. The paper's structure is challenging to follow, with concepts introduced before being fully explained and circular references between ideas. The writing style is dense and often obscures rather than clarifies the underlying concepts. Many paragraphs contain multiple complex ideas without sufficient development of each.

3) **Lack of clear methodology section**: The paper lacks a clear methodological section. It presents theoretical constructs without explaining how they were derived or how they could be validated. For a theory paper, there should be clearer explanation of the analytical methods used to develop the framework.

4) **Limited empirical support**: The limited examples provided (particularly regarding the Periodic Table and Standard Model) are interesting but insufficiently developed to demonstrate the framework's utility. The paper would benefit from detailed case studies showing how the theory addresses specific problems in AI or information theory.

5) **Unclear practical implications**: While claiming to provide "first principles" for AI, the concrete applications and implications for AI system development remain underspecified.

### **Summary**:
The paper addresses an important problem and presents interesting ideas about the nature of meaning and information that could potentially inform AI development. However, in its current form, it lacks the clarity, precision, and supporting evidence needed for a significant contribution to the field

---

### Official Review · Reviewer_YQEj · 2025-03-05

**Rating:** 6
**Confidence:** 4

**Review:**

This paper introduces a framework for a theory of meaning based on information theory and entropy, addressing a gap identified by Claude Shannon and Warren Weaver. It argues that current views on information lack a structured way to define meaning and proposes a dualistic Subject-Object (S-O) model to bridge this gap. The author examines Shannon’s Signal Entropy and extends it into a multi-level framework that describes general intelligence, adaptive processes, and structured information. The paper suggests that meaning emerges from trial-and-error interactions with the physical world, leading to hierarchical patterns of intelligence. By integrating entropy, semiotics, and evolutionary processes, this approach seeks to provide a foundation for artificial and human intelligence, offering a general framework for understanding how information becomes meaningful.

Strengths of the Paper:
1. The paper addresses a fundamental gap in information theory by focusing on meaning.
2. The paper incorporates entropy and semiotics, linking physical and informational processes.
3. The paper provides a cross-disciplinary perspective, unifying theories from physics, biology, and AI.
4. Suggests a testable framework for building intelligent systems with meaningful interactions.

Weaknesses of the Paper:
1. Lack of empirical validation, relying mostly on theoretical arguments.
2. Definitions of meaning and intelligence remain broad, requiring further refinement.

---

### Decision · Program_Chairs · 2025-03-05

Accept (Poster)